# Clinical and molecular epidemiology of influenza viruses from Romanian patients hospitalized during the 2019/20 season

Victor Daniel Miron[1]☉, Leontina Bănică[2]☉, Oana Săndulescu[1,2], Simona Paraschiv[1,2]*, Marius Surleac[2,3], Dragoş Florea[1,2], Ovidiu Vlaicu[2], Petre Milu[1,2], Anca Streinu-Cercel[1,2], Anuta Bilaşco[2], Dan Oţelea[2], Daniela Piţigoi[1,2], Adrian Streinu-Cercel[1,2], Anca Cristina Drăgănescu[2]

1 Carol Davila University of Medicine and Pharmacy, Bucharest, Romania, 2 "Prof. Dr. Matei Bals" National Institute for Infectious Diseases, Bucharest, Romania, 3 Research Institute of the University of Bucharest (ICUB), Bucharest, Romania

☉ These authors contributed equally to this work.
* simona.paraschiv@umfcd.ro

**Data Availability Statement:** All relevant data are within the manuscript and its Supporting Information files.

## Abstract

Two main mechanisms contribute to the continuous evolution of influenza viruses: accumulation of mutations in the hemagglutinin and neuraminidase genes (antigenic drift) and genetic re-assorts (antigenic shift). Epidemiological surveillance is important in identifying new genetic variants of influenza viruses with potentially increased pathogenicity and transmissibility. In order to characterize the 2019/20 influenza epidemic in Romania, 1042 respiratory samples were collected from consecutive patients hospitalized with acute respiratory infections in the National Institute for Infectious Diseases "Prof. Dr. Matei Balş", Bucharest Romania and tested for influenza A virus, influenza B virus and respiratory syncytial virus (RSV) by real-time PCR. Out of them, 516 cases were positive for influenza, with relatively equal distribution of influenza A and B. Two patients had influenza A and B co-infection and 8 patients had influenza-RSV co-infection. The most severe cases, requiring supplemental oxygen administration or intensive care, and the most deaths were reported in patients aged 65 years and over. Subtyping showed the predominance of A(H3N2) compared to A(H1N1)pdm09 pdm09 (60.4% and 39.6% of all subtyped influenza A isolates, respectively), and the circulation of Victoria B lineage only. Influenza B started to circulate first (week 47/2019), with influenza A appearing slightly later (week 50/2019), followed by continued co-circulation of A and B viruses throughout the season. Sixty-eight samples, selected to cover the entire influenza season and all circulating viral types, were analysed by next generation sequencing (NGS). All A(H1N1)pdm09 sequences identified during this season in Romania were clustered in the 6b1.A clade (sub-clades: 6b1.A.183P -5a and 6b1.A.187A). For most A(H1N1)pdm09 sequences, the dominant epitope was Sb ($p_{epitope}$ = 0.25), reducing the vaccine efficacy by approximately 60%. According to phylogenetic analysis, influenza A(H3N2) strains circulating in this season belonged predominantly to clade 3C.3A, with only few sequences in clade 3C.2A1b. These 3C.2A1b sequences, two of which belonged to vaccinated patients, harbored mutations in antigenic sites leading to potential reduction of vaccine efficacy. Phylogenetic analysis of influenza B, lineage

**Funding:** The study received funding from the Global Influenza Hospital Surveillance Network (GIHSN) project and the Development of Robust and Innovative Vaccine Effectiveness (DRIVE) project, as follows: GIHSN project was co-funded by the Foundation for Influenza Epidemiology; the DRIVE study has received support from the EU/EFPIA Innovative Medicines Initiative 2 Joint Undertaking (DRIVE, grant n° 777363). Both studies were co-funded by the National Institute for Infectious Diseases "Prof. Dr. Matei Balş", Bucharest, Romania. GIHSN and DRIVE contributed to study design but had no role in data collection and analysis, decision to publish, or preparation of the manuscript. The National Institute for Infectious Diseases „Prof. Dr. Matei Balş" contributed to study design, data collection and analysis, but had no role in the decision to publish, or preparation of the manuscript. The authors VDM, LB, OS, SP, MS, DF, OV, Anca S-C, AB, DO, DP, Adrian S-C and ACD were supported by GIHSN project. MS was supported by Research Institute of the University of Bucharest (ICUB) grant no. 20964/30.10.2020. The funder ICUB provided support in the form of fellowship for the author MS, but did not have any additional role in the study design, data collection and analysis, decision to publish, or preparation of the manuscript.

**Competing interests:** OS, Anca S-C, and Adrian S-C report being investigators in influenza clinical trials by Shionogi and F. Hoffmann-La Roche, outside the scope of the submitted work. No other authors have competing interests to declare.

Victoria, sequences showed that the circulating strains belonged to clade V1A3. As compared to the other viral types, fewer mutations were observed in B/Victoria strains, with limited impact on vaccine efficiency based on estimations.

## Introduction

Due to constant viral evolution, influenza strains included in the vaccines are revised yearly based on WHO's recommendations [1]. Therefore, epidemiological surveillance and, in particular, molecular characterization of the circulating viral strains, are essential for making informed predictions on the upcoming influenza circulation and for defining the vaccine composition.

All known pandemics were generated by influenza A through genetic re-assortments of different viral strains followed by human transmission. However, random viral mutations can also increase the viral pathogenicity. In contrast with type A viruses, influenza B strains were reported almost exclusively in humans. Two distinct lineages (Victoria and Yamagata) circulated continuously starting with the 1980s [2]. In recent years, in human populations, A(H1N1)pdm09 and A(H3N2) subtypes co-circulated with B type viruses, one or both lineages being present at a given moment.

Hemagglutinin (HA) represents the main target for neutralizing antibodies and therefore, is essential for vaccine design. Five antigenic sites have been described in the globular head domain (HA1) of A(H1N1)pdm09 strains (Sa, Sb, Ca1, Ca2 and Cb) [3] and A(H3N2) (A to E) [4]. Four antigenic sites have been identified for influenza B viruses, located in the 120 loop, 150 loop, 160 loop and 190 helix [5]. Mutations at these sites may compromise vaccine efficiency and should be continuously monitored.

Surveillance data on influenza are gathered systematically through international consortia such as Global Influenza Hospital Surveillance Network (GIHSN) and Development of Robust and Innovative Vaccine Effectiveness (DRIVE) [6, 7]. Romania participates in these international networks by collecting yearly prospective data from patients hospitalized for severe acute respiratory infections. During the season 2019/20, the National Institute for Infectious Diseases "Prof. Dr. Matei Balş" has also performed whole genome sequencing (WGS) for a number of samples in addition to collecting standardized clinical and virological data. Only a limited number of complete genetic sequences of influenza are available for the Central-Eastern European region. Phylogenetic analysis performed on the HA encoding segment has been clearly demonstrated to be an important contributor in improving the understanding of the dynamics of the epidemic within a season, in specific geographical regions [5, 8]. In this context, the aim of the current study was to characterize the influenza epidemic based on the data from patients hospitalized for this indication in a tertiary care hospital from this Central-Eastern European country.

## Methods

### Study population

The full study protocol has been previously described [6, 7, 9]. In brief, all consecutive patients hospitalized in the National Institute for Infectious Diseases "Prof. Dr. Matei Balş", a reference hospital for infectious diseases in Bucharest Romania, for a recent onset of influenza-like illness/ severe acute respiratory infection (ILI/SARI) during the influenza surveillance period were tested by RT-PCR for influenza and respiratory syncytial virus (RSV), and a complete medical history was collected, along with clinical and outcome data of the respective hospital

admission. The study started on 11 November 2019, at the same time with the national SARI surveillance, and was stopped on 22 March 2020, earlier than initially planned, because of the COVID-19 pandemic, which prompted the conversion of the institute into a COVID-19-only hospital. Sets of two respiratory swabs were collected on viral transport media (VTM) (COPAN ITALIA S.P.A., Brescia, Italy) for all the patients that fulfilled the selection criteria.

## Real time RT-PCR for influenza diagnosis and subtyping

Detection of influenza A, influenza B and RSV viruses was performed by real-time RT-PCR using the GenXpert instrument (Cepheid, Sunnyvale, CA, USA) and the Allplex™ Respiratory Panel 1 kit (Seegene, Seul, South Coreea). Genetic subtyping of samples positive for Influenza B was done by an inhouse assay as previously described [9].

## Next generation sequencing (NGS)

Eighty-five influenza positive samples were randomly selected from all epidemiological weeks included in the study, to undergo WGS. The RNA was extracted from 500 μl VTM using QiAmp DSP Virus (Qiagen, Hilden, Germany), followed by reverse transcription and PCR amplification using Superscript III OneStep RT-PCR (ThermoFisher Scientific, Waltham, Massachusetts, USA) and primer sets previously described [10, 11]. The amplicons were purified using AMPure DNA beads (Agilent Santa Clara, CA, USA) and quantified with Qubit HS DNA kit (Thermo Scientific). Nextera DNA Flex Library Prep Kit (Illumina, San Diego, CA, USA) was used for library preparation according to the manufacturer recommendations. Before sequencing, the quality and quantity of DNA pool libraries were verified using fluorescence reagents compatible with the 2100 Bioanalyser (Agilent, Santa Clara, CA, USA) and the Qubit 4 Fluorimeter (Thermo Fisher Scientific). 68 samples were successfully sequenced on the MiSeq platform (Illumina) by using the paired-end shotgun strategy and MiSeq reagent kit v.3 (600 cycles). The NGS assembly method followed a double approach protocol like the one previously described [12] in which the final consensus sequences are a result of *de novo* and corresponding reference mapping runs. For mapping purposes, we used one reference sequence for each viral type.

## Phylogenetic analysis, reference sequences

The generated consensus sequences were aligned against the reference sequences retrieved from the GISAID database, which were selected as follows: other available sequences from Romania during the 2019–2020 influenza season, sequences of the influenza strains recommended for the 2019/2020 vaccine and clade defining sequences indicated by Nextstrain platform (https://nextstrain.org/flu/seasonal). The final sequence alignments were generated using MEGA 7 and then the resulting alignments were used as input for the generation of phylogenetic trees using the Geneious Tree Builder method with the following algorithm: HKY genetic distance model, Neighbor-Joining tree build method, 1000 replicates bootstrap. The final trees were visualized and annotated using FigTree v1.4.4. The datasets used in the phylogenetic analysis are available as S1–S4 Datasets. The first three contains the HA nucleotide alignments used to generate the phylogenetic trees for Influenza A(H1N1)pdm09, A(H3N2) and B respectively. S4 Dataset contains the NA nucleotide sequences analysed in this study.

## Nucleotide sequences accession number

The sequences reported in this study were deposited in GISAID databases under the accession numbers: EPI_ISL 403939–403961; 413462–413484; 445192–445212, 470881.

## Mutation detection, prediction of N-glycosylation sites and structural overview

The Fluserver tool (https://flusurver.bii.a-star.edu.sg/) was used to identify mutations in the analyzed sequences. The nucleotide alignments were uploaded and compared with the most similar sequence included in 2019/20 vaccine.

The sequences (amino acids) reported here were aligned with the most similar strains included in the vaccine and evaluated for potential N-glycosylation sites using NetNGlyc server [13].

The impact of mutations from a structural point of view was analyzed by mapping them on the X-ray crystal structures of the influenza HA. The 3D structures used here were the ones suggested by the Fluserver predictions and then downloaded from the PDB database (https://www.rcsb.org): 3M6S; 4WE8 and 4FQK.

## Prediction of vaccine efficacy

To evaluate the divergence of the influenza strains circulating in Romania during the 2019/20 influenza season and the strain included in the vaccine, the corresponding protein sequences were aligned with the most similar vaccine strain and mutations in B cell antigenic epitopes were identified. The antigenic differences were used to predict vaccine efficacy using the $P_{epitope}$ model, as previously reported [5]. The $P_{epitope}$ for each antigenic site was calculated as the ratio between the number of mutated amino acids and the total number of amino acids in the epitope. The dominant epitope was used to predict the vaccine efficacy based on the formula: $E = -2.47 \times P_{epitope} + 0.47$ for A(H3N2), $E = -1.19 \times P_{epitope} + 0.53$ for A(H1N1)pdm09 and $E = -0.864 \times P_{epitope} + 0.6824$ for Influenza B [5].

## Statistical analysis

Categorical data are presented as frequency/percentage and compared using the Chi-squared test. For continuous variables, data distribution was checked using the Shapiro-Wilk test; for non-normally distributed variables, median and interquartile range (IQR) are reported, the Mann-Whitney U test was used for statistical comparisons between the study groups and the Kruskal-Wallis H test was used to compare more than two independent groups. The statistical analysis was performed in IBM SPSS Statistics for Windows, version 25 (IBM Corp, Armonk, NY, USA), with a p value <0.05 being considered statistically significant.

## Ethics statement

The study was approved by the Bioethics Committee of the National Institute for Infectious Diseases „Prof. Dr. Matei Balş" (approval numbers: 9550/2019 and 9379/2019). All patients, or their appropriate legal representatives, provided written informed consent prior to performing any study-related procedures.

## Results

### Study population

A total of 1162 patients admitted for ILI or SARI were screened, and 1042 patients met the inclusion criteria and were tested by RT-PCR for the presence of influenza viruses and RSV (Fig 1). There was a balanced distribution by gender (51.4% male, n = 536), with a median age of 8.8 years (IQR:2.3, 32.9 years). From the overall study group, approximately one third of the patients (30.6%, n = 319) had at least one chronic condition, the most common being

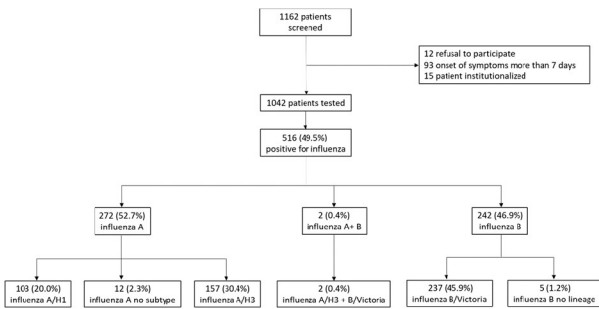

**Fig 1. Study design flowchart.** Percentages of influenza types and subtypes were calculated relative to the total number of influenza positive cases.

cardiovascular disease (n = 142, 13.6% overall), which was most often present in young adults (n = 53/263, 20.2%) and in elderly adults (n = 82/100, 82.0%). The median length of hospital stay was 5.5 days (IQR: 3.5, 7), with an intensive care unit (ICU) admission rate of 6.0% (n = 62) and a case-fatality rate of 0.8% (n = 8). The vaccination rate for all patients included in the study was 6.4% (n = 67). Vaccination rates were slightly different, but not statistically significant, in the influenza positive and negative groups (5.6% and 7.2%, respectively).

## Circulation of viral subtypes/lineages in the 2019/20 season

We identified a relatively high influenza positivity rate (49.5%) and a balanced circulation of influenza A and B viruses: of 516 positive cases, 272 (52.7%) were influenza A, 242 (46.9%) were influenza B, and co-infection with both viruses was identified in 2 cases (0.4%) (Fig 1). More than 95% of the influenza A strains were subtyped, showing the predominance of A/H3 compared to A/H1 (60.4% and 39.6% of all subtyped influenza A isolates, respectively). For most of the influenza B cases (lineage determination was performed for 97.9% of B cases), genotyping data indicated the circulation of Victoria lineage only (Fig 1).

The first case of influenza was identified in week 47/2019, and the last positive case was detected in week 11/2020 (Fig 2A). The highest rate of positivity was recorded in weeks 5-7/2020. Among 67 vaccinated patients, 29 were confirmed with influenza, 21 of which with influenza A (6-A/H1, 12-A/H3) and 8 with influenza B (7-B/Victoria). The positivity rate was not significantly different in vaccinated versus non-vaccinated patients (43.2% vs 49.9%, p = 0.313). Fig 2B illustrates the distribution by age and viral subtype and lineage, showing that among influenza positive cases A/H1 predominated mostly in adults aged 50 to 70 years, followed by patients with ages 85 and over, and by the group of children with ages below 10 years old. By comparison, A/H3 was dominant among younger study participants (40 to 50 years old and 65 to 75 years old), and in adolescents and young adults (15 to 25 years old), while B/Victoria predominated in children of all ages and young adults up to 40 years old, with a smaller peak in adults aged 75 and over (Fig 2B).

## Characteristics of cases of laboratory-confirmed influenza (LCI)

Children tested positive for influenza significantly more frequently than adults (58% vs 33.6%, p<0.001). Moreover, about three quarters of confirmed cases were attributed to paediatric patients (76.4%, n = 394), therefore the median age of the positive group for influenza was 6.2 years (IQR:2.2, 17.0 years).

Clinical symptoms varied with age. Overall, fever was the predominant symptom in both children (98.2%) and adults (95.9%) (p = 0.166), but children associated more frequently:

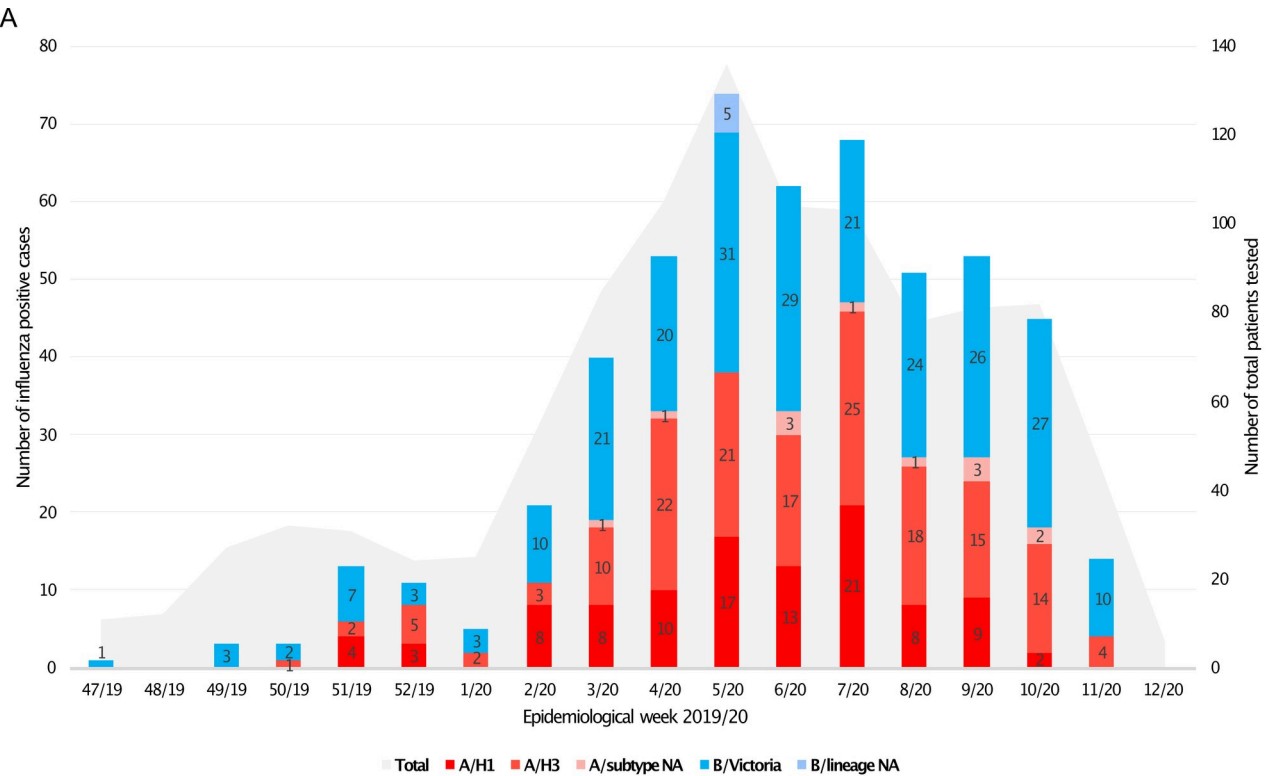

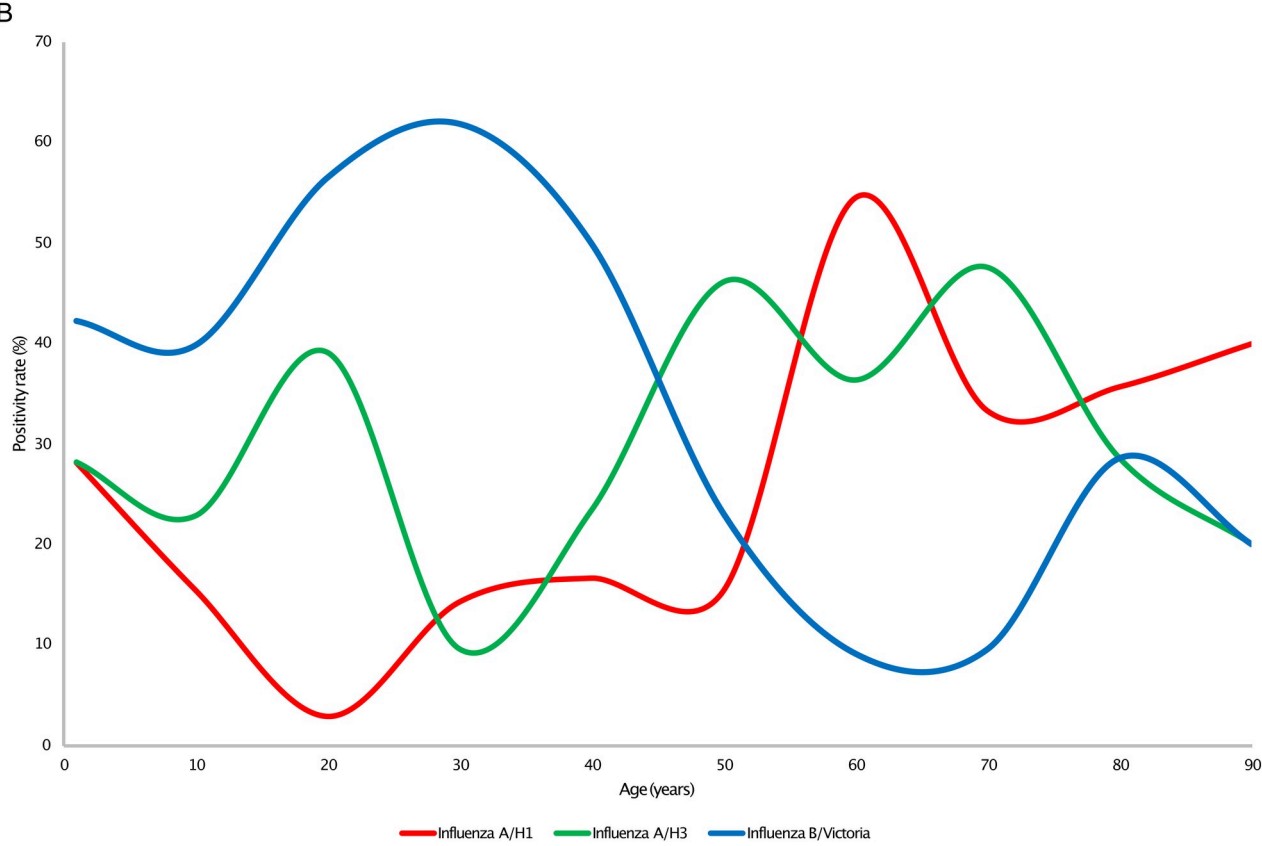

**Fig 2. A**. Weekly distribution of influenza positive cases by viral types, subtypes/lineages during the influenza season 2019/20. **B.** Distribution of attack rates for different influenza types, by patient age. Data is expressed as percentage of the laboratory-confirmed influenza cases.

cough (97.0% vs. 82%, p<0.001, OR = 3.4, 95%CI: 1.5–7.9) and nasal congestion (93.7% vs. 50.0%, p<0.001, OR = 14.8, 95%CI:8.6–25.3). The presentation of the clinical picture by age groups is summarized in Table 1.

The rate of admission to the ICU was equal between children and adults (4.1% for each, p = 0.889), but the latter experienced respiratory failure and required supplemental oxygen more often compared to children (19.7% vs. 7.1%, p<0.001, OR = 3.2, 95%CI: 1.8–5.8). The pre-existence of a chronic disease among the adult population was associated with a 4.1-fold increased risk of respiratory failure (p = 0.011, OR = 4.1, 95%CI: 1.3–12.8).

When analyzing specifically the data for elderly patients (65 years and older), 89.7% (n = 26) of them had pre-existing comorbidities, and 44.8% (n = 13) developed respiratory failure requiring supplemental oxygen administration, a significantly higher percentage compared with younger adults (11.8%, n = 11, p<0.001, OR = 6.1, 95%CI:2.3–15.9)–Table 1.

**Table 1. Characteristics of patients with laboratory-confirmed influenza by age group.**

| Characteristics | Infants, <1 year, n = 71 | Toddlers, 1–2 years, n = 92 | Preschoolers, 3–4 years, n = 71 | School children, 5–13 years, n = 137 | Teenagers, 14–17 years, n = 23 | Adults, 18–64 years, n = 93 | Elderly adults, ≥65 years, n = 29 | Total patients, n = 516 |
|---|---|---|---|---|---|---|---|---|
| Male gender, n (%) | 44 (62.0%) | 62 (67.4%) | 38 (53.5%) | 80 (58.4%) | 13 (56.5%) | 33 (35.5%) | 9 (31.0%) | 279 (54.1%) |
| Clinical characteristic | | | | | | | | |
| Fever, n (%) | 69 (97.2%) | 92 (100%) | 71 (100%) | 133 (97.1%) | 22 (95.7%) | 90 (96.8%) | 27 (93.1%) | 504 (97.7%) |
| Malaise, n (%) | 49 (69.0%) | 74 (80.4%) | 66 (93.0%) | 122 (89.1%) | 19 (82.6%) | 77 (93.5%) | 24 (82.8%) | 431 (83.5%) |
| Headache, n (%) | N/A | 8 (8.7%) | 31 (43.7%) | 93 (67.9%) | 18 (78.3%) | 69 (74.2%) | 15 (51.7%) | 234 (45.3%) |
| Myalgia, n (%) | N/A | 4 (4.3%) | 24 (33.8%) | 59 (43.1%) | 13 (56.5%) | 62 (66.7%) | 12 (41.4%) | 174 (33.7%) |
| Cough, n (%) | 70 (98.6%) | 90 (97.8%) | 70 (98.6%) | 131 (95.6%) | 21 (91.3%) | 58 (62.4%) | 9 (31.0%) | 492 (95.3%) |
| Sore throat, n (%) | N/A | 19 (20.7%) | 39 (54.9%) | 101 (73.7%) | 15 (65.2%) | 58 (62.4%) | 9 (31.0%) | 241 (46.7%) |
| Dyspnea, n (%) | 6 (8.5%) | 8 (8.7%) | 3 (4.2%) | 7 (5.1%) | 3 (13.0%) | 29 (31.2%) | 16 (55.2%) | 72 (14.0%) |
| Nasal congestion, n (%) | 70 (98.6%) | 88 (95.7%) | 71 (100%) | 124 (90.5%) | 16 (69.6%) | 48 (51.6%) | 13 (44.8%) | 430 (83.3%) |
| Deterioration, n (%) | 8 (11.3%) | 30 (32.6%) | 29 (40.8%) | 57 (41.6%) | 12 (52.2%) | 40 (43.0%) | 19 (65.5%) | 195 (37.8%) |
| Chronic conditions, n (%) | 6 (8.5%) | 9 (9.8%) | 10 (14.1%) | 18 (13.1%) | 6 (26.1%) | 48 (51.6%) | 26 (89.7%) | 123 (23.8%) |
| Cardiovascular disease, n (%) | 0 | 0 | 0 | 3 (2.2%) | 2 (8.7%) | 20 (21.5%) | 23 (79.3%) | 48 (9.3%) |
| COPD, n (%) | 0 | 0 | 0 | 0 | 0 | 2 (2.2%) | 0 | 2 (0.4%) |
| Asthma, n (%) | 0 | 0 | 1 (1.4%) | 2 (1.5%) | 1 (4.3%) | 2 (2.2%) | 2 (6.9%) | 8 (1.6%) |
| Diabetes mellitus, n (%) | 0 | 0 | 0 | 0 | 0 | 13 (14.0%) | 7 (24.1%) | 20 (3.9%) |
| Renal impairment, n (%) | 1 (1.4%) | 0 | 0 | 1 (0.7%) | 0 | 3 (3.2%) | 5 (17.2%) | 10 (1.9%) |
| Rheumatologic disease, n (%) | 0 | 0 | 0 | 0 | 1 (4.3%) | 6 (6.5%) | 4 (13.8%) | 11 (2.1%) |
| Neurological disease, n (%) | 0 | 6 (6.5%) | 4 (5.6%) | 7 (5.1%) | 2 (8.7%) | 3 (3.2%) | 5 (17.2%) | 27 (5.2%) |
| Liver disease, n (%) | 0 | 0 | 0 | 0 | 0 | 10 (10.8%) | 6 (20.7%) | 16 (3.1%) |
| Neoplasm, n (%) | 0 | 1 (1.1%) | 0 | 2 (1.5%) | 0 | 3 (3.2%) | 6 (20.7%) | 12 (2.3%) |
| Obesity, n (%) | N/A | N/A | 0 | 2 (1.5%) | 2 (8.7%) | 14 (15.1%) | 4 (13.8%) | 22 (4.3%) |
| HIV infection, n (%) | 1 (1.4%) | 1 (1.1%) | 0 | 0 | 0 | 7 (7.5%) | 0 | 9 (1.7%) |
| Days of hospitalization, median (IQR) | 5 (3, 6) | 5 (3, 6) | 4 (3, 5) | 4 (3, 5) | 5 (5, 7) | 4 (3, 7) | 8 (7, 15) | 5 (3, 6) |
| Respiratory failure with supplemental oxygen, n (%) | 4 (5.6%) | 9 (9.8%) | 4 (5.6%) | 10 (7.3%) | 1 (4.3%) | 11 (11.8%) | 13 (44.8%) | 52 (10.1%) |
| ICU admission, n (%) | 3 (4.2%) | 4 (4.3%) | 2 (2.8%) | 6 (4.4%) | 1 (4.3%) | 2 (2.2%) | 3 (10.3%) | 21 (4.1%) |
| Deaths, n (%) | 0 | 0 | 0 | 0 | 0 | 1 (1.1%) | 2 (6.9%) | 3 (0.6%) |

COPD—chronic obstructive pulmonary disease, ICU–intensive care unit, N/A–not applicable

Three deaths occurred in patients with LCI, all in adults, unvaccinated against influenza, who had severe influenza requiring management in the ICU: a 69-year-old man with cardiovascular disease and diabetes, positive for influenza A (subtype not available), a 64-year-old woman with cardiovascular disease, chronic kidney disease, diabetes, rheumatic disease and leukemia and an 81-year-old woman with cardiovascular disease, asthma, kidney disease and neurological disease—both positive for influenza A/H1.

## Comparison of influenza type distribution in cases of LCI

When analyzing the cases based on the type of influenza virus, we did not identify differences in terms of gender or median age of patients. However, influenza A was significantly more frequent in elderly patients (79.3% of them) (p = 0.003, OR = 3.6, 95%CI:1.4–9.0) while B viruses circulated predominantly in younger patients (94.6% of cases in patients under 40 years). Fig 2B illustrates the attack rate for influenza types (proportion of the total positive cases) in different age categories. There were no differences in clinical presentation, evolution, ICU admission rate, or death rate between the two influenza groups (S1 Table).

## Co-infections in patients with LCI

Two cases of co-infection A/H3 and B/Victoria influenza were identified, both in children: a 14-year-old girl with pre-existing rheumatic disease, who required hospitalization for 6 days, and an 8-year-old boy, without chronic diseases, who required 11 days of hospitalization; both cases had not been vaccinated against influenza in the respective season and had favourable evolution under etiologic treatment.

Eight patients with LCI presented co-infection with RSV: 3 cases with A/H1 and 5 cases with B/Victoria. Two of them occurred in adults (30 years and 63 years old, respectively), the rest in children (ages 1–17). The presence of RSV co-infection increased the duration of hospitalization to 8.0±6.2 days vs. 5.2±3.4 days, p = 0.021, without other significant changes in the clinical course.

## Influenza A(H1N1)pdm09 genetic characterization

HA and NA are under immune selective pressure and evolve rapidly, being frequently used for phylogenetic analysis purposes. [14]. In this study, both HA and NA genes were used for phylogenetic classification, returning similar results. Phylogenetic analysis of NA genes is presented in S1 Fig.

H1N1 sequences reported in this study (n = 18) were classified as belonging to 1.A.3.3.2 clade using swine H1 clade classification in Influenza Research Database (https://www.fludb.org/brc/h1CladeClassifier.spg). Further analysis using phylogenetic tools and clade defining sequences (https://nextstrain.org/flu/seasonal) showed that 4/18 belonged to 6b1.A.183P -5a clade (highlighted in blue in Fig 3), while the other sequences were assigned to 6b1.A.187A clade (marked in yellow in the tree). In both clades, other sequences reported elsewhere in Romania during the 2019/20 season clustered together with the sequences reported in this study. An average of 9 amino acid mutations per HA sequence was found in the Romanian A(H1N1)pdm09 strains. Several mutations were found in antigenic sites: L161I, S164T in Sa, T185I, D187A, Q189E in Sb, A139T in Ca2. The variability analysis of H1N1 sequences at antigenic sites showed Sb to be the dominant epitope, with T185I, D187A and Q189E found in 14 of 18 analyzed H1N1 strains circulating in Romania during the 2019/20 season. The predicted vaccine efficacy estimated using the $p_{epitope}$ model [5] against the A(H1N1)pdm09 influenza strain included in the 2019/20 vaccine (A/Brisbane/02/2018) showed a potential vaccine efficacy of 44% (E = 23.3% of 53%, $p_{epitope}$ = 0).

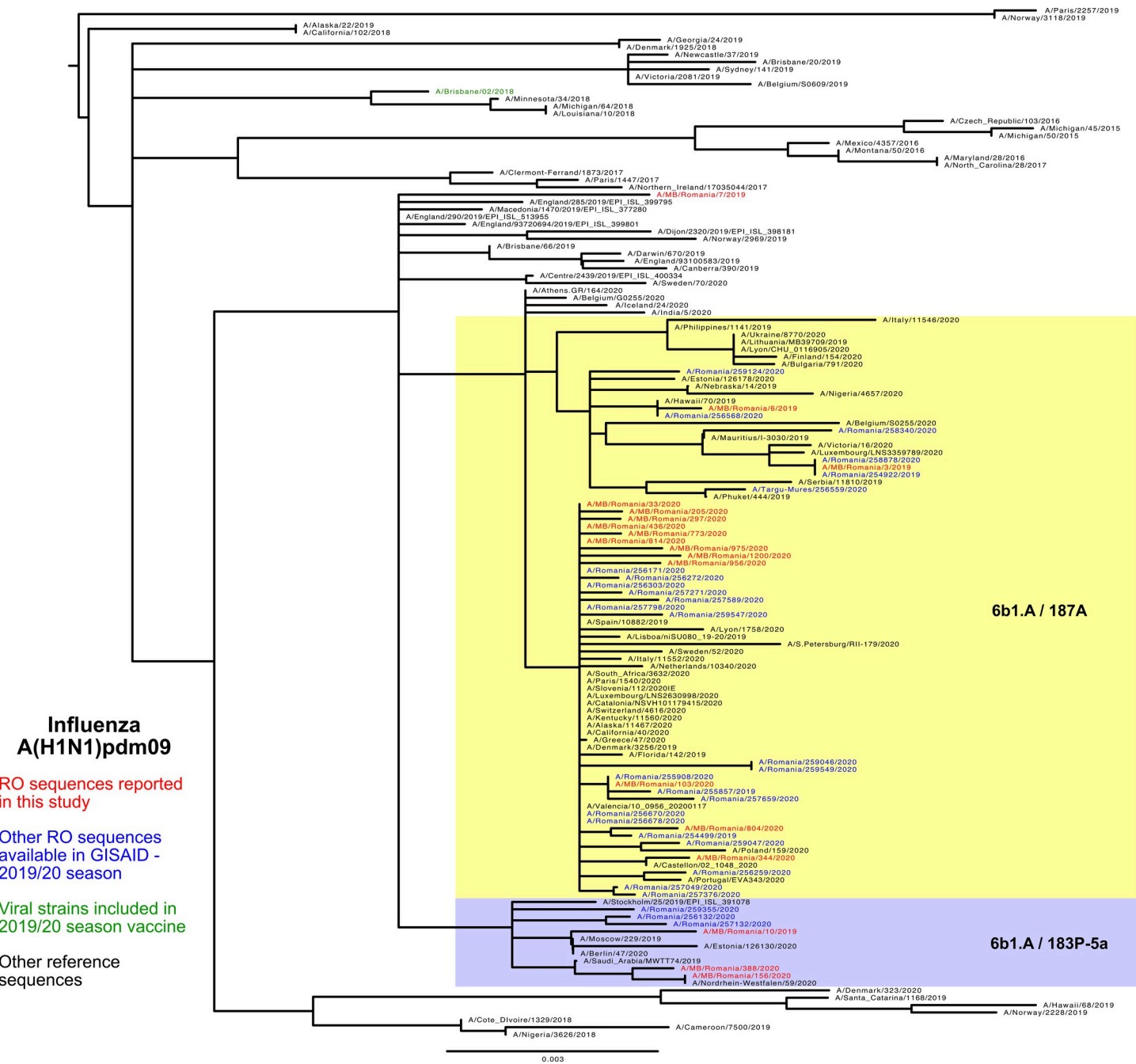

**Fig 3. Phylogenetic analysis of A(H1N1)pdm09 hemagglutinin sequences.**

From a structural point of view, there were seven amino acid mutations found in all HA sequences reported in this study: G45R, N129D, T185I, R223Q, N260D, A282P, V298I when compared to A/Brisbane/02/2018 reference sequence. Other mutations were observed with different frequencies in the analyzed sequences (e.g T25A, A139T, D187A, Q189E, Y230C). Only one HA sequence had a mutation (T25A) affecting the N-glycosylation process. However, this modification was outside the antigenic sites.

## Influenza A(H3N2) genetic characterization

We have observed the segregation of Romanian influenza A(H3N2) strains into clades 3C3A and 3C2A (Fig 4A) when using clade defining sequences as controls. Seventeen sequences belong to clade 3C3A and clustered together with most of the other Romanian strains isolated during the 2019/20 influenza season and also with the strain included in the Northern hemisphere vaccine (A/Kansas/14/2017). This clade is highlighted in yellow in Fig 4. According to phylogenetic analysis, four of the A(H3N2) sequences were classified as clade 3C2A1b and clustered together with the strain included in the 2020 vaccine for the Southern hemisphere (A/South Australia/34/2019). This cluster is highlighted in blue in Fig 4. Two of the patients with 3C2A1b strains had been vaccinated; the other two were epidemiologically connected and not vaccinated. The sequences of the latter are identical, clustering together with sequences from Moldova, Czech Republic and Netherlands in the phylogenetic tree. Very few 3C2A1b strains were reported

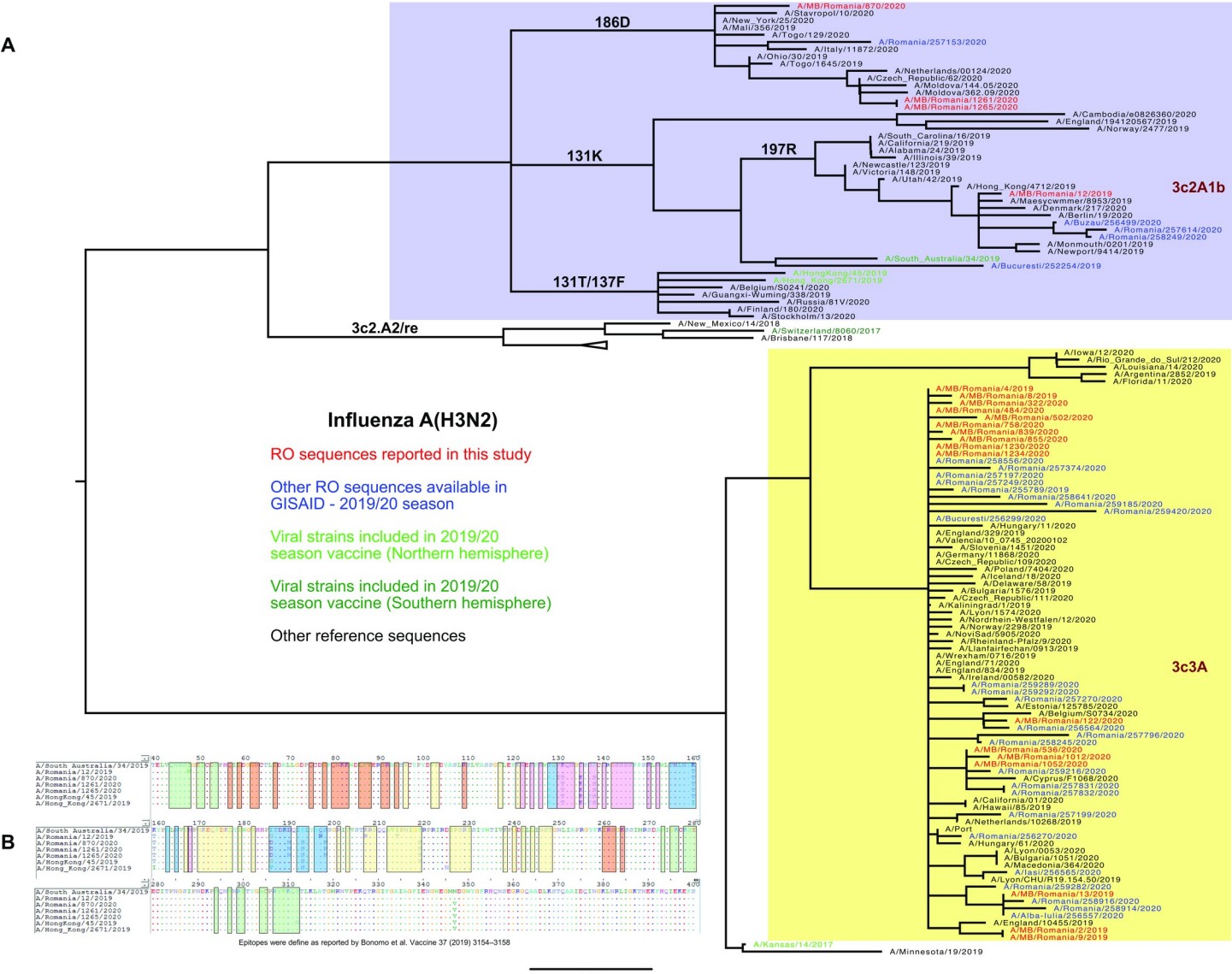

**Fig 4. A.** Phylogenetic analysis of A(H3N2) hemagglutinin sequences. **B.** Mutations in antigenic sites identified in HA nucleotide sequences compared with vaccine strains.

during 2019/20 season in Romania, with only five other Romanian sequences clustering together in the 3C2A1b group. This sub-clade was reported with higher prevalence in countries from the Southern hemisphere (Australia, New Zeeland, African countries).

One of these four 3C2A1b strains (infecting a vaccinated patient) was characterized by the presence of 131K and 197R mutations; in this subclade, three other sequences reported elsewhere in Romania were present. The other three 3C2A1b sequences had additional 135K and 186D mutations and were classified as subclade 3C2A1b/186D.

The T135K substitution, present in 3C2A1b/186D sequences, is associated with loss of one N-glycosylation site (N133) in antigenic site A. This has an impact on the immune response, making this antigenic site more accessible to the neutralizing antibodies.

The analysis of hemagglutinin 3C3A sequences showed limited evolution when antigenic sites were compared to the strain included in the 2019/20 influenza vaccine (A/Kansas/14/2017), K82R in epitope E was identified in 7 samples. In contrast, 3C2A1b HA sequences presented an increased number of mutations as compared to the most similar influenza vaccine strain (A/South Australia/34/2019). These sequences showed the highest variability in the antigenic site B; T128A, K160T, I186D, D190N, F193S, S198P for subclade 3C2A1b/186D and K160T, I186G, Q197R for subclade 3C2A1b/197R (Fig 4B). Calculating the impact of these mutations on vaccine efficacy using $p_{epitope}$ model, an important loss was suggested when subclade 3C2A1b/197R sequence was analysed (E = 12% of 47%, $p_{epitope}$ = 0). Similarly, a potential drop-out in vaccine efficacy was suggested when subclade 3C2A1b/186D sequences were analysed.

## Influenza B / Victoria genetic characterization

Phylogenetic analysis of Influenza B lineage Victoria sequences showed that the strains isolated in Romania during the 2019/2020 season belonged to clade V1A3, together with the sequence included in the vaccine for season 2020–2021 in both the Northern and Southern hemispheres (B/Washington/2/2019). The strain included in the vaccine for the 2019/2020 season in the Northern hemisphere was B/Colorado/06/2018 that belongs to a different clade, V1A1 (Fig 5). A hallmark for V1A3 strains is the 162–164 triple deletion in HA1, while V1A1 sequences may have a 162–163 double deletion.

As compared to the strain included in the 2019/2020 Influenza vaccine, all 29 samples reported in this study presented the following substitutions: G129D, K136E, 164del, V178I, T195N and K496R. Fifteen sequences had an additional N126K mutation in the 120 loop antigenic site, while the other fourteen sequences presented E128K and G133R mutations in the same antigenic site. Calculating the impact of mutations in the 120 loop antigenic site on vaccine efficacy using $p_{epitope}$ model, a small decrease was seen: E = 83% (N126K mutation), respectively E = 77% (E128K and G133R mutations). Particular mutations, present in several sequences, are of interest. Among these, E196G (found in 11 samples in 190-helix) was found to be associated with increased likelihood of glycosylated 195N (NGTQ, score = 0.72, jury agreement 9/9 ++) and might protect the virus from the neutralizing antibodies.

## Discussion

Romania participates in international respiratory infection surveillance networks with annual prospective data collection. Only a limited number of complete genetic sequences of influenza are available for Central-Eastern European region. The 2019/20 influenza season started and finished earlier than in previous years, potentially due to travel restrictions and sanitary protection measures in the context of the COVID-19 pandemic.In the studied season, the first influenza cases in Romania were identified in week 47/2019, initially as sporadic circulation of influenza B; influenza A cases appeared slightly later, starting with week 50/2019 [15]. The last

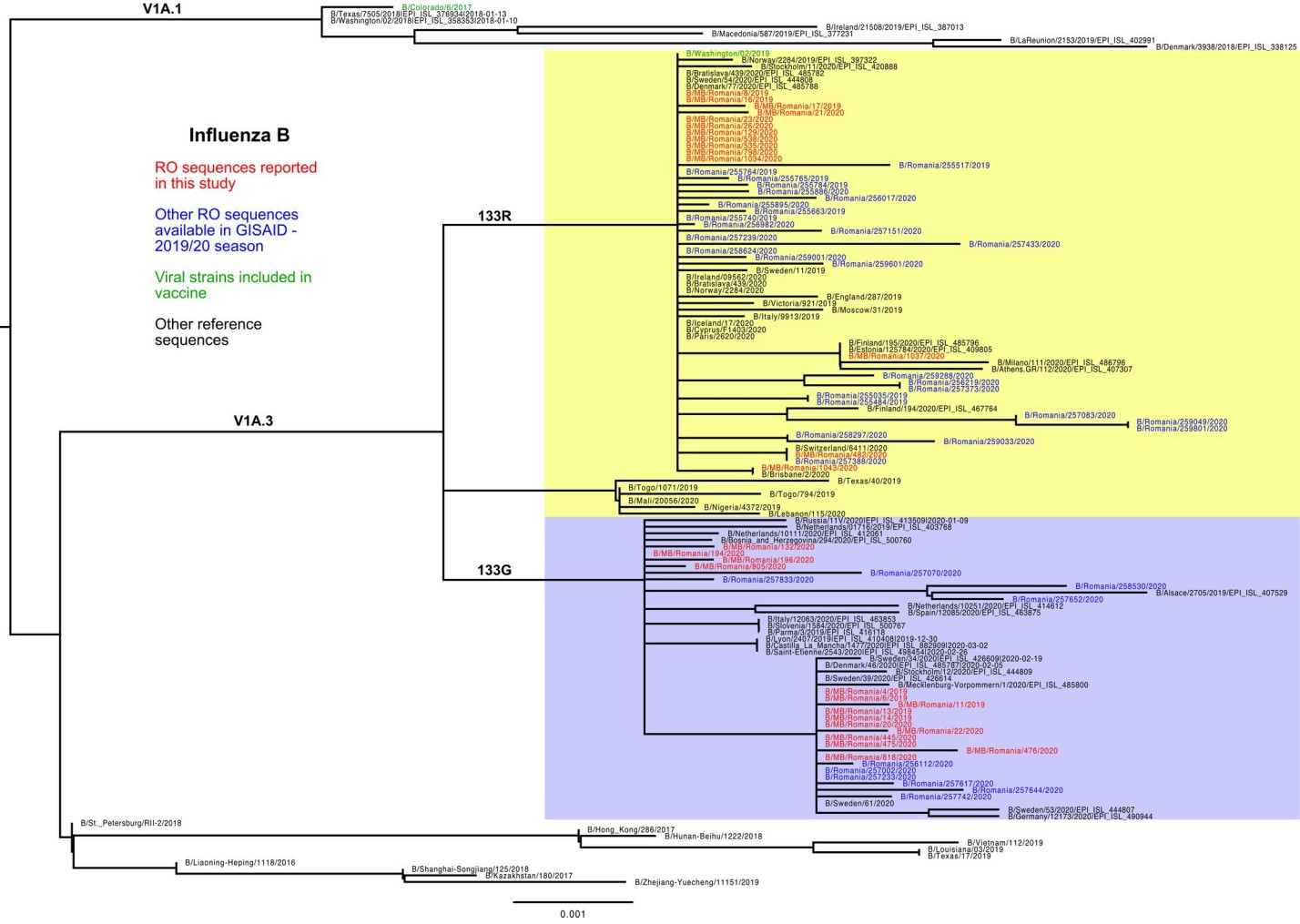

**Fig 5. Phylogenetic analysis of B/Victoria hemagglutinin sequences.**

confirmed cases occurred in week 20/2020 in Europe and week 11/2020 in Romania, earlier than in previous years, most probably due to social distancing measures prompted by the COVID-19 pandemic: the initial national lockdown started in week 12/2020. This is in line with data communicated by the European Centre for Disease Prevention and Control [15], which reported that influenza activity mostly returned to baseline levels during week 13/2020 throughout Europe.

Influenza A/H1, A/H3 and B/Victoria were detected during the influenza season in Romania with slightly different viral type distribution compared to data reported for the European region where 66% of the registered cases were type A, with similar proportion of A/H1 and A/H3 subtypes (51 vs. 49%) [15]. In our study, type B viruses accounted for 46% of the total infections. B/Victoria viruses were more frequently observed in young patients under 40 years. Most severe cases and deaths were reported in A/H1 infections and patients over 65 years of age. In this regard, Romania is not dissimilar to other European countries [15].

Overall, influenza affected all age groups, but adults (18 years and over) had a 3.2-fold higher risk of respiratory failure requiring supplemental oxygen. Among adults, in elderly patients (65 years and older) this risk was 6.1-fold higher compared to younger adults. This age-dependent

increase in risk of influenza severity or complications leading to prolonged hospitalization had been previously described in our patient population [16, 17] and elsewhere [18], and it has remained valid for subsequent influenza seasons, including the one reported here.

In our population, we have seen a predominance of pediatric patients among cases of LCI. While this may reflect to a certain extent a real increased circulation of influenza viruses among young children, who are known to be a risk group for influenza, this may also be driven at least in part by an increased addressability of pediatric patients with ILI to tertiary care rather than primary care services in our country, as previously discussed [19, 20]. Furthermore, a potential propensity towards managing fever and administering antiviral treatment in children in the hospital rather than as outpatients cannot be ruled out, and this is probably reflected in the rather short 5 days median length of hospitalization in infants and toddlers. However, children remain an important risk group for influenza, and vaccination coverage rates in children remain suboptimal in Europe: historically reported median vaccine coverage is 10.9% [21].

During the 2019/20 influenza season the A(H3N2) viruses circulating in the European region belonged to clades 3C.3a and 3C.2a, subgroup 3C.2a1b (54 and 46%, respectively) [15]. The majority of A(H3N2) strains reported in this study belonged to 3C.3a clade and few strains were 3C.2A1b (either 3C.2A1b/186D or 3C.2A1b/197R). The phylogenetic analysis indicated that Romanian 3C.3a sequences clustered with the reference vaccine sequence A/Kansas/14/2017, whereas 3C.2a1b sequences were highly divergent from it and more similar to A/South-Australia/34/2019. This strain was first included in the 2020 vaccine for the Southern hemisphere. Still, a high number of mutations were observed in Romanian 3C.2a1b sequences (antigenic site B) when compared with A/South-Australia/34/2019 vaccine sequence. These mutations were predicted to have a strong impact on vaccine efficacy: up to a 75% decrease for 3C2A1b/197R strains and complete loss for 3C2A1b/186D analyzed strains. The 2020/21 vaccine in Northern hemisphere covered 3C.2a1b strains by including A/Hong Kong/2671/2019 or A/Hong Kong/45/2019 (belong to 3C.2a1b/137F). The A(H3N2) strain included in 2020/21 vaccine covers most of the mutations observed in the antigenic sites of 3C.2a1b sequences.

The decision to include a 3C.3a strain in the 2019/20 vaccine was debatable since most of the strains circulating during the previous season in Europe and other parts of the world were 3C.2a. Gouma S and collaborators showed that sera from ferrets and humans immunized with 3C.3a strains had a good neutralizing activity against 3C.3a strains, but not against 3C.2a. Antigenic site B is responsible for this poor immune response [22].

The 190 position is one of the conserved positions in the antigenic site B and RBS of H3N2 HA and mutations in this position can affect receptor binding and viral replication fitness [23]. Important differences were observed at this position when sequences from the two clades were analyzed: 190D in 3C.3a, 190N in 3C.2a1b.

Among the mutations identified in antigenic sites for 3C2A1b sequences, K160T is of particular interest: in addition to diminishing virus neutralization by specific antibodies generated in response to vaccination creating a new potential N-glycosylation in position 158 thus masking the site from neutralizing antibodies [22]. However, the score does not reach significant levels (score < 0.7). Removal of N158 glycosylation at the HA globular head in H5N1 strain (A/Vietnam/1203/2004) was reported to increase binding affinity, use of both α-2,3SAL and α2,6SAL residues, resulting in more efficient viral replication and an increased antibody response in ferrets [24].

Most A(H1N1)pdm09 strains circulating in Romania, as identified in our study, belonged to 6b1.A.183P -5a clade, and few were assigned as 6b1.A.187A. For the majority of A(H1N1)pdm09 sequences, the dominant epitope was Sb ($p_{epitope}$ = 0.25); the predicted impact of mutations on the vaccine efficacy was of approximately 60% reduction. Most of the mutations reside on the exposed receptor binding domain (RBD), but the Y230C mutation was found at

the interface of the monomers. The Y230C mutation was found in only one sample from this study (A/Romania/975/2020/EPI_ISL_445196) together with another mutation which may be of interest–T25A.

Limited viral variability was observed when analyzing the B/Victoria strains circulating during the flu 2019/20 season in Romania as compared to the vaccine strain for the corresponding season (B/Colorado/06/2017). The mutations identified were estimated to have little impact on vaccine efficacy. In addition to this observation, only few vaccinated patients positive for influenza (8/29) were infected with B strains.

Previous studies showed that mutations in HA and NA genes confer resistance to both antivirals and antibodies. Mutations like N203V in HA and E329K in NA were associated with broad neutralizing antibodies viral escape [25]. In this study, the mutation analysis of NA sequences showed limited evolution as compared with the vaccine strains. None of the analyzed NA sequences presented the H275Y mutation associated with oseltamivir resistance (S4 Dataset).

This study has a set of strengths; the data described here are derived from systematic prospective surveillance for ILI/SARI, as part of large international influenza networks. However, the current study also has certain limitations. The data is collected from a single tertiary care center and cannot be generalized to the country as a whole. Furthermore, an involuntary selection bias driven by hospital admission of more severe cases in our institute, which is a national reference center for infectious diseases, cannot be excluded.

In conclusion, we have described the co-circulation of A/H3, A/H1 and B Victoria viruses in the 2019/20 season in patients hospitalized for SARI in a reference center for infectious diseases in Bucharest, Romania, disproportionately affecting the youngest and eldest age groups in terms of attack rate and severity, respectively. Phylogenetic analysis indicated the circulation of different viral variants, some of them harboring mutations in HA gene with an important impact on vaccine efficiency.

## Supporting information

**S1 Table. Characteristics of patients depending on the type of influenza virus (A or B).**
(DOCX)

**S1 Fig. Phylogenetic analysis of neuraminidase sequences.** A. Influenza A(H1N1)pdm09. B. Influenza A(H3N2). C. Influenza B.
(TIFF)

**S1 Dataset. HA nucleotide alignment used in the phylogenetic analysis of Influenza A (H1N1)pdm09 sequences.**
(FAS)

**S2 Dataset. HA sequence alignment used to generate the phylogenetic tree of Influenza A (H3N2).**
(FAS)

**S3 Dataset. HA nucleotide alignment used in the phylogenetic analysis of Influenza B/Victoria sequences.**
(FAS)

**S4 Dataset. Influenza NA nucleotide sequences analysed in this study.**
(FASTA)

## Acknowledgments

The authors thank all study participants and the hospital staff for their involvement in this project.

## Author Contributions

**Conceptualization:** Oana Săndulescu, Anca Streinu-Cercel, Anuta Bilaşco, Dan Oţelea, Daniela Piţigoi, Adrian Streinu-Cercel, Anca Cristina Drăgănescu.

**Data curation:** Victor Daniel Miron, Leontina Bănică, Oana Săndulescu, Simona Paraschiv, Dragoş Florea, Petre Milu, Anca Streinu-Cercel, Anuta Bilaşco, Daniela Piţigoi, Anca Cristina Drăgănescu.

**Formal analysis:** Victor Daniel Miron, Leontina Bănică, Simona Paraschiv, Marius Surleac, Dragoş Florea, Ovidiu Vlaicu, Petre Milu.

**Funding acquisition:** Oana Săndulescu, Anca Streinu-Cercel, Daniela Piţigoi, Anca Cristina Drăgănescu.

**Investigation:** Victor Daniel Miron, Leontina Bănică, Oana Săndulescu, Simona Paraschiv, Marius Surleac, Dragoş Florea, Ovidiu Vlaicu, Anca Streinu-Cercel, Anuta Bilaşco, Daniela Piţigoi, Adrian Streinu-Cercel, Anca Cristina Drăgănescu.

**Methodology:** Leontina Bănică, Oana Săndulescu, Simona Paraschiv, Marius Surleac, Dragoş Florea, Anca Streinu-Cercel, Anuta Bilaşco, Daniela Piţigoi, Adrian Streinu-Cercel, Anca Cristina Drăgănescu.

**Software:** Leontina Bănică, Marius Surleac.

**Supervision:** Simona Paraschiv, Dan Oţelea.

**Validation:** Leontina Bănică, Simona Paraschiv, Marius Surleac, Dragoş Florea.

**Visualization:** Leontina Bănică, Simona Paraschiv, Marius Surleac, Ovidiu Vlaicu.

**Writing – original draft:** Victor Daniel Miron, Leontina Bănică, Simona Paraschiv, Marius Surleac, Dragoş Florea, Ovidiu Vlaicu, Petre Milu, Dan Oţelea.

**Writing – review & editing:** Victor Daniel Miron, Leontina Bănică, Oana Săndulescu, Simona Paraschiv, Marius Surleac, Dragoş Florea, Petre Milu, Anca Streinu-Cercel, Anuta Bilaşco, Dan Oţelea, Daniela Piţigoi, Adrian Streinu-Cercel, Anca Cristina Drăgănescu.

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
