## [Decision Letter · Decision Letter 0]

2 Sep 2021

PONE-D-21-14205

Clinical and molecular epidemiology of influenza viruses from Romanian patients hospitalized during the 2019/20 season

PLOS ONE

Dear Dr. Paraschiv,

Thank you for submitting your manuscript to PLOS ONE. After careful consideration, we feel that it has merit but does not fully meet PLOS ONE’s publication criteria as it currently stands. Therefore, we invite you to submit a revised version of the manuscript that addresses the points raised during the review process.

We look forward to receiving your revised manuscript.

Kind regards,

Ahmed S. Abdel-Moneim, Ph.D.

Academic Editor

PLOS ONE

Journal Requirements:

 "No. The funders had no role in study design, data collection and analysis, decision to publish, or preparation of the manuscript."

"The GIHSN project was co-funded by the Foundation for Influenza Epidemiology and National Institute for Infectious Diseases „Prof. Dr. Matei Balș”, Bucharest, Romania. The DRIVE study has received support from the EU/EFPIA Innovative Medicines Initiative 2 Joint Undertaking (DRIVE, grant n° 777363) and was co-funded by National Institute for Infectious Diseases „Prof. Dr. Matei Balș”, Bucharest, Romania. Marius Surleac was supported by Research Institute of the University of Bucharest (ICUB) grant no. 20964/30.10.2020. The study was supported by POSCCE program CRCBABI project (642/2014)."

"The GIHSN project was co-funded by the Foundation for Influenza Epidemiology and National Institute for Infectious Diseases „Prof. Dr. Matei Balș”, Bucharest, Romania. The DRIVE study has received support from the EU/EFPIA Innovative Medicines Initiative 2 Joint Undertaking (DRIVE, grant n° 777363) and was co-funded by National Institute for Infectious Diseases „Prof. Dr. Matei Balș”, Bucharest, Romania. Marius Surleac was supported by Research Institute of the University of Bucharest (ICUB) grant no. 20964/30.10.2020. The study was supported by POSCCE program CRCBABI project (642/2014)."

"No. The funders had no role in study design, data collection and analysis, decision to publish, or preparation of the manuscript."

We note that one or more of the authors is affiliated with the funding organization, indicating the funder may have had some role in the design, data collection, analysis or preparation of your manuscript for publication; in other words, the funder played an indirect role through the participation of the co-authors. If the funding organization did not play a role in the study design, data collection and analysis, decision to publish, or preparation of the manuscript and only provided financial support in the form of authors' salaries and/or research materials, please do the following:

a. Review your statements relating to the author contributions, and ensure you have specifically and accurately indicated the role(s) that these authors had in your study. These amendments should be made in the online form.

b. Confirm in your cover letter that you agree with the following statement, and we will change the online submission form on your behalf: 

“The funder provided support in the form of salaries for authors [insert relevant initials], but did not have any additional role in the study design, data collection and analysis, decision to publish, or preparation of the manuscript. The specific roles of these authors are articulated in the ‘author contributions’ section.

7. We note that you have included the phrase “data not shown” in your manuscript. Unfortunately, this does not meet our data sharing requirements. PLOS does not permit references to inaccessible data. We require that authors provide all relevant data within the paper, Supporting Information files, or in an acceptable, public repository. Please add a citation to support this phrase or upload the data that corresponds with these findings to a stable repository (such as Figshare or Dryad) and provide and URLs, DOIs, or accession numbers that may be used to access these data. Or, if the data are not a core part of the research being presented in your study, we ask that you remove the phrase that refers to these data.

8. Please upload a new copy of Figure 2 as the detail is not clear. Please follow the link for more information: " ext-link-type="uri" xlink:type="simple">https://blogs.plos.org/plos/2019/06/looking-good-tips-for-creating-your-plos-figures-graphics/"
" ext-link-type="uri" xlink:type="simple">https://blogs.plos.org/plos/2019/06/looking-good-tips-for-creating-your-plos-figures-graphics/".

9. Please include captions for your Supporting Information files at the end of your manuscript, and update any in-text citations to match accordingly. Please see our Supporting Information guidelines for more information: http://journals.plos.org/plosone/s/supporting-information

Reviewers' comments:

Reviewer's Responses to Questions

**Comments to the Author**

Review Comments to the Author

Reviewer #1: The study presents a comprehensive overview on the molecular characteristics of influenza viruses circulating in Romania during the 2019/20 season. The authors describe the viruses under consideration of different earmarks and hypothesize about the clinical or epidemiological impact of their findings. The data is thoroughly described and presented clearly and well written.

Some minor remarks:

- Please apply the international rules for influenza A subtype nomenclature �A(H1N1)pdm09 and A(H3N2) instead of A/H1N1/pdm09 and A/H3N2

- The citations in paragraph 3 of the introduction (lines 62-67) seem imprecise. Please check for publications focusing on the topic of interest

- Lines 149-150: Please explain or cite the origin of the applied formulas

Reviewer #2: The manuscript entitled “Clinical and molecular epidemiology of influenza viruses from Romanian patients hospitalized during the 2019/20 season” by Miron V.D. et al. presents an observational study based on the data from influenza surveillance in the National Institute for Infectious Diseases.

Major point:

Manuscript contains repetitive information and can be significantly shortened.

Comments:

Abstract.

- lines 25-29. “Romania… pandemic” should be moved to Discussion

- lines 31-32. Name the hospital, spell out RSV

- line 35. “Increased number of severe cases and death… compared to …”

- lines 37-38. “… circulate first (week 47/2019), ….later (50/2019)…..

- line 40. Start description of phylogenetic analysis from H1N1, then H3N2, and then B virus.

Introduction.

- Omit generalized unnecessary sentences (like line 51 etc.).

- line 78. “… current study..”

- line 93. …IAV, IBV and RSV

Results.

Move Figures’ titles and legends out of this section.

- lines 168 -171. With a median age of 8.8 years, the most common chronic condition being cardiovascular disease seems very strange.

- lines 173-174. Differences in vaccination rates should be statistically confirmed or omit “…slightly different…”.

- line 179. “…. (0.4%) (Fig.1)”.

- line 180. I do not understand where 60.3% and 39.6% came from?

- Fig. 1 is confusing. It should more clearly indicate that % subtype was calculated from all positive influenza cases.

- line 184. “…week 11/2020 (Fig. 2A).”

- Omit sentence “A complete…”

- Description of patients by age (shown in Fig. 2B) is missing.

- lines 195-229. Description of the data in the paragraph and data shown in Table 1 do not correlate with each other.

- lines 251-252. Show NA phylogeny in a Supplementary figure.

- Consider a Supplementary Table showing all HA and NA mutations

- lines 303-305. Repetitive information

---

## [Author Response · Author response to Decision Letter 0]

17 Sep 2021

Reviewer comments Authors responses

Reviewer #1

The study presents a comprehensive overview on the molecular characteristics of influenza viruses circulating in Romania during the 2019/20 season. The authors describe the viruses under consideration of different earmarks and hypothesize about the clinical or epidemiological impact of their findings. The data is thoroughly described and presented clearly and well written. 

Some minor remarks:

- Please apply the international rules for influenza A subtype nomenclature �A(H1N1)pdm09 and A(H3N2) instead of A/H1N1/pdm09 and A/H3N2

 R:Thank you for the appreciative comments and suggestions. We have revised the whole text as recommended.

- The citations in paragraph 3 of the introduction (lines 62-67) seem imprecise. Please check for publications focusing on the topic of interest

 R: Thank you for the suggestion. We have replace the reference 3 (doi: 10.1128/JVI.01381-14 ) and reference 4 (doi: 10.1371/journal.pone.0041895)

- Lines 149-150: Please explain or cite the origin of the applied formulas

 R: We have added the reference as suggested (reference 5)

Reviewer #2

The manuscript entitled “Clinical and molecular epidemiology of influenza viruses from Romanian patients hospitalized during the 2019/20 season” by Miron V.D. et al. presents an observational study based on the data from influenza surveillance in the National Institute for Infectious Diseases.

Major point:

Manuscript contains repetitive information and can be significantly shortened.

 R:Thank you for the suggestion. We have revised the article to remove any redundancies, and we have also removed information that you marked as too general or unnecessary. All changes can be seen in the tracked version of the manuscript. By removing redundancies, we have shortened the manuscript, however, in order to respond to other observations mentioned by the reviewers below, we also had to add some new paragraphs, i.e., distribution by age and viral subtype of positive cases, etc. 

Comments:

Abstract.

- lines 25-29. “Romania… pandemic” should be moved to Discussion

 R: We have modified the text accordingly. 

- lines 31-32. Name the hospital, spell out RSV

 R: Done and done.

- line 35. “Increased number of severe cases and death… compared to …”

 R: Thank you for the clarification request. We have revised the phrase for clarity in line with the study’s results: “The most severe cases, requiring supplemental oxygen administration or intensive care, and the most deaths were reported in patients aged 65 years and over.”

- lines 37-38. “… circulate first (week 47/2019), ….later (50/2019)…..

 R: We revised the phrase according to your indication: “Influenza B started to circulate first (week 47/2019), with influenza A appearing slightly later (week 50/2019)”.

- line 40. Start description of phylogenetic analysis from H1N1, then H3N2, and then B virus.

 R: We have modified the text accordingly.

Introduction.

- Omit generalized unnecessary sentences (like line 51 etc.).

 R: Done

- line 78. “… current study..”

 R: Revised as instructed: “the aim of the current study was”

- line 93. …IAV, IBV and RSV Because we did not use the “IAV” and “IBV” abbreviations anywhere throughout the manuscript, we kept the full spelling of the virus names here, except for RSV, which is used multiple times in its abbreviated form. Thank you for understanding. 

Results.

Move Figures’ titles and legends out of this section.

 R: We have followed the PLOS ONE's style requirements.

- lines 168 -171. With a median age of 8.8 years, the most common chronic condition being cardiovascular disease seems very strange.

 R: Thank you for the comment. Indeed, comorbidities in general and cardiovascular disease in particular were mostly present in adults. We have now rephrased to clarify this in the text:

“There was a balanced distribution by gender (51.4% male, n=536), with a median age of 8.8 years (IQR:2.3, 32.9 years). From the overall study group, approximately one third of the patients (30.6%, n=319) had at least one chronic condition, the most common being cardiovascular disease (n=142, 13.6% overall), which was most often present in young adults (n=53/263, 20.2%) and in elderly adults (n=82/100, 82.0%)”

- lines 173-174. Differences in vaccination rates should be statistically confirmed or omit “…slightly different…”.

 R: We have revised phrasing to clarify that the difference was not statistically significant: “Vaccination rates were slightly different, but not statistically significant, in the influenza positive and negative groups (5.6% and 7.2%, respectively).”

- line 179. “…. (0.4%) (Fig.1)”.

 R: Figure 1 is now cited here in the text, according to your suggestion.

- line 180. I do not understand where 60.3% and 39.6% came from?

 R: We have now clarified in the text that these represented percentages from among all subtyped influenza strains: “showing the predominance of A/H3 compared to A/H1 (60.4% and 39.6% of all subtyped influenza A isolates, respectively)”

- Fig. 1 is confusing. It should more clearly indicate that % subtype was calculated from all positive influenza cases.

 R: We have added the following table legend, for clarification of how the percentages were calculated: “Figure 1 legend: Percentages of influenza types and subtypes were calculated relative to the total number of influenza positive cases”.

- line 184. “…week 11/2020 (Fig. 2A).”

- Omit sentence “A complete…”

 R: We added the citation to the figure to the phrase, as instructed, and we deleted the phrase stating with “A complete…”

- Description of patients by age (shown in Fig. 2B) is missing.

 R: We have now added the description of viral subtype distribution by age: “Figure 2 B illustrates the distribution by age and viral subtype and lineage, showing that among influenza positive cases A/H1 predominated mostly in adults aged 50 to 70 years, followed by patients with ages 85 and over, and by the group of children with ages below 10 years old. By comparison, A/H3 was dominant among younger study participants (40 to 50 years old and 65 to 75 years old), and in adolescents and young adults (15 to 25 years old), while B/Victoria predominated in children of all ages and young adults up to 40 years old, with a smaller peak in adults aged 75 and over (Figure 2 B).”

- lines 195-229. Description of the data in the paragraph and data shown in Table 1 do not correlate with each other.

 R: We agree that the description of the data is somewhat different in the text than in the table in this portion of the manuscript. To a certain degree, this was intentional, since we wanted to avoid having the text only repeat the information presented in the Table, and because we felt that it would overcomplicate the manuscript’s flow if we discussed in the text the particularities of each smaller age group presented in Table 1.

Therefore, for illustration in the text we chose to only report the comparisons performed between children and adults, which in our opinion clearly demarcate two different and heterogeneous patient populations whose particularities should be well understood and recognized in the clinic. By mistake, there were indeed two citations to Table 1 where the data was not exactly presented by age group, and we have now removed these and we have added a citation in the correct place, in the paragraph that accurately describes the data for elderly patients, in complete accordance with how it is also reported in Table 1. 

- lines 251-252. Show NA phylogeny in a Supplementary figure.

 R: Thank you for your suggestion, we have added the NA phylogenetic tree in S1 Fig.

- Consider a Supplementary Table showing all HA and NA mutations

 R: Most of the relevant information is already available in figure 4 and extra data are available in S1-S4 datasets. This considered, an extra table seems to us redundant.

- lines 303-305. Repetitive information 

R: Thank you for observation, we have modified the text accordingly.

Journal requirements

 Authors responses

 1.R: We have checked the PLOS ONE's style requirements and modified the submission where it was needed 

 2.R: The reference list was revised and modified as suggested by the reviewers (see reference 3 and 4)

 "No. The funders had no role in study design, data collection and analysis, decision to publish, or preparation of the manuscript."

3. R: Please update the funding statement as following:

The study received funding from the Global Influenza Hospital Surveillance Network (GIHSN) project and the Development of Robust and Innovative Vaccine Effectiveness (DRIVE) project, as follows:

GIHSN project was co-funded by the Foundation for Influenza Epidemiology; the DRIVE study has received support from the EU/EFPIA Innovative Medicines Initiative 2 Joint Undertaking (DRIVE, grant n° 777363). Both studies were co-funded by the National Institute for Infectious Diseases „Prof. Dr. Matei Balș”, Bucharest, Romania.

GIHSN and DRIVE contributed to study design but had no role in data collection and analysis, decision to publish, or preparation of the manuscript. The National Institute for Infectious Diseases „Prof. Dr. Matei Balș” contributed to study design, data collection and analysis, but had no role in the decision to publish, or preparation of the manuscript. The authors VDM, LB, OS, SP, MS, DF, OV, ASC, AB, DO, DP, ASC and ACD were supported by GIHSN project. Marius Surleac was supported by Research Institute of the University of Bucharest (ICUB) grant no. 20964/30.10.2020. The funder ICUB provided support in the form of fellowship for author [MS], but did not have any additional role in the study design, data collection and analysis, decision to publish, or preparation of the manuscript. 

"The GIHSN project was co-funded by the Foundation for Influenza Epidemiology and National Institute for Infectious Diseases „Prof. Dr. Matei Balș”, Bucharest, Romania. The DRIVE study has received support from the EU/EFPIA Innovative Medicines Initiative 2 Joint Undertaking (DRIVE, grant n° 777363) and was co-funded by National Institute for Infectious Diseases „Prof. Dr. Matei Balș”, Bucharest, Romania. Marius Surleac was supported by Research Institute of the University of Bucharest (ICUB) grant no. 20964/30.10.2020. The study was supported by POSCCE program CRCBABI project (642/2014)."

"The GIHSN project was co-funded by the Foundation for Influenza Epidemiology and National Institute for Infectious Diseases „Prof. Dr. Matei Balș”, Bucharest, Romania. The DRIVE study has received support from the EU/EFPIA Innovative Medicines Initiative 2 Joint Undertaking (DRIVE, grant n° 777363) and was co-funded by National Institute for Infectious Diseases „Prof. Dr. Matei Balș”, Bucharest, Romania. Marius Surleac was supported by Research Institute of the University of Bucharest (ICUB) grant no. 20964/30.10.2020. The study was supported by POSCCE program CRCBABI project (642/2014)."

 4.R: We have reviewed the Acknowledgments as following:

Acknowledgments :

The authors thank all study participants and the hospital staff for their involvement in this project. 

"No. The funders had no role in study design, data collection and analysis, decision to publish, or preparation of the manuscript."

We note that one or more of the authors is affiliated with the funding organization, indicating the funder may have had some role in the design, data collection, analysis or preparation of your manuscript for publication; in other words, the funder played an indirect role through the participation of the co-authors. If the funding organization did not play a role in the study design, data collection and analysis, decision to publish, or preparation of the manuscript and only provided financial support in the form of authors' salaries and/or research materials, please do the following:

a. Review your statements relating to the author contributions, and ensure you have specifically and accurately indicated the role(s) that these authors had in your study. These amendments should be made in the online form.

b. Confirm in your cover letter that you agree with the following statement, and we will change the online submission form on your behalf: 

“The funder provided support in the form of salaries for authors [insert relevant initials], but did not have any additional role in the study design, data collection and analysis, decision to publish, or preparation of the manuscript. The specific roles of these authors are articulated in the ‘author contributions’ section.

 5.a) R: The specific roles of these authors were articulated in the ‘author contributions’ section:

Author contributions

Victor Daniel Miron: Formal analysis, Investigation, Data Curation, Writing - Original Draft, Writing - Review Editing, Visualization

Leontina Bănică: Conceptualization, Methodology, Investigation, Data Curation, Phylogenetic analysis, Writing - Review Editing 

Oana Săndulescu: Conceptualization, Methodology, Investigation, Data Curation, Writing - Review Editing

Simona Paraschiv: Conceptualization, Methodology, Investigation, Data Curation, Phylogenetic analysis, Writing - Review Editing, Data submission

Marius Surleac: Conceptualization, Methodology, Investigation, Data Curation, Phylogenetic analysis, Writing - Review Editing

Dragoș Florea: Conceptualization, Methodology, Investigation, Data Curation, Writing - Review Editing

Ovidiu Vlaicu: Methodology, Investigation, Data Curation, Phylogenetic analysis, Writing Editing

Petre Milu: Methodology, Investigation, Data Curation, Phylogenetic analysis, Writing Editing

Anca Streinu-Cercel: Conceptualization, Methodology, Investigation, Data Curation, Writing - Review Editing

Anuta Bilașco: Conceptualization, Methodology, Investigation, Data Curation, Writing - Review Editing

Dan Oțelea: Conceptualization, Methodology, Investigation, Data Curation, Writing - Review Editing

Daniela Pițigoi: Conceptualization, Methodology, Investigation, Data Curation, Writing - Review Editing

Adrian Streinu-Cercel: Conceptualization, Methodology, Investigation, Data Curation, Writing - Review Editing

Anca Cristina Drăgănescu: Conceptualization, Methodology, Investigation, Data Curation, Writing - Review Editing

5.b) R: We have reviewed the financial disclosure (see below). Marius Surleac has added his new affiliation: “Research Institute of the University of Bucharest (ICUB), Bucharest, Romania” and added the following statement:

“The funder ICUB provided support in the form of fellowship for author [MS], but did not have any additional role in the study design, data collection and analysis, decision to publish, or preparation of the manuscript.

 6. R: Data Availability statement should be updated as following:

Four datasets have been uploaded as supplementary files: S1 Dataset, S2 Dataset, S3 Dataset contains the HA nucleotide alignments used to generate the phylogenetic trees for Influenza A(H1N1)pdm09, A(H3N2) and B respectively. S4 Dataset contains the NA nucleotide sequences analysed in this study.

We have also revised the Phylogenetic analysis, reference sequences section accordingly.

7. We note that you have included the phrase “data not shown” in your manuscript. Unfortunately, this does not meet our data sharing requirements. PLOS does not permit references to inaccessible data. We require that authors provide all relevant data within the paper, Supporting Information files, or in an acceptable, public repository. Please add a citation to support this phrase or upload the data that corresponds with these findings to a stable repository (such as Figshare or Dryad) and provide and URLs, DOIs, or accession numbers that may be used to access these data. Or, if the data are not a core part of the research being presented in your study, we ask that you remove the phrase that refers to these data.

 7. R: To respond your demand, we have included S4 Dataset and modified the text accordingly.

8. Please upload a new copy of Figure 2 as the detail is not clear. Please follow the link for more information: https://blogs.plos.org/plos/2019/06/looking-good-tips-for-creating-your-plos-figures-graphics/" https://blogs.plos.org/plos/2019/06/looking-good-tips-for-creating-your-plos-figures-graphics/".

 8. R: A new copy of Figure 2 has been uploaded.

9. Please include captions for your Supporting Information files at the end of your manuscript, and update any in-text citations to match accordingly. Please see our Supporting Information guidelines for more information: http://journals.plos.org/plosone/s/supporting-information

9. R: We have included the captions for your Supporting Information files at the end of the manuscript, and update any in-text citations to match accordingly.

---

## [Decision Letter · Decision Letter 1]

6 Oct 2021

Clinical and molecular epidemiology of influenza viruses from Romanian patients hospitalized during the 2019/20 season

PONE-D-21-14205R1

Dear Dr. Paraschiv,

We’re pleased to inform you that your manuscript has been judged scientifically suitable for publication and will be formally accepted for publication once it meets all outstanding technical requirements.

Kind regards,

Ahmed S. Abdel-Moneim, Ph.D.

Academic Editor

PLOS ONE

---

## [Editor Report · Acceptance letter]

3 Nov 2021

PONE-D-21-14205R1 

Clinical and molecular epidemiology of influenza viruses from Romanian patients hospitalized during the 2019/20 season 

Dear Dr. Paraschiv:

I'm pleased to inform you that your manuscript has been deemed suitable for publication in PLOS ONE. Congratulations! Your manuscript is now with our production department. 

Kind regards, 

on behalf of

Prof. Ahmed S. Abdel-Moneim 

Academic Editor

PLOS ONE